# Growth and Carcass Characteristics of Beef-Cross-Dairy-Breed Heifers and Steers Born to Different Dam Breeds

**DOI:** 10.3390/ani12070864

**Published:** 2022-03-29

**Authors:** Holly R. Williamson, Nicola M. Schreurs, Stephen T. Morris, Rebecca E. Hickson

**Affiliations:** Animal Science, School of Agriculture and Environment, Massey University, Palmerston North 4442, New Zealand; n.m.schreurs@massey.ac.nz (N.M.S.); s.t.morris@massey.ac.nz (S.T.M.); r.hickson@massey.ac.nz (R.E.H.)

**Keywords:** dam, genetics, dairy-beef, growth rate, carcass

## Abstract

**Simple Summary:**

The dairy industry is an important source of animals for the beef industry in New Zealand. Since 1998, the proportion of Friesian–Jersey cows in dairy herds has increased from 19% to 49%. The aim of this study was to investigate the effect of dam breed on growth and carcass characteristics of beef-cross-dairy-breed progeny born to Friesian, Friesian-cross, Friesian–Jersey and Jersey-cross cows. Dam breed had an effect on all growth traits, as well as carcass weight, fat colour, fat depth, and ossification score. An increase in Jersey genetics in calves retained for beef finishing would result in calves that take a longer time to reach weaning weight and have a lighter carcass and yellower fat.

**Abstract:**

Approximately two thirds of the annual beef kill in New Zealand originates from the dairy industry. The recent increase in Jersey genetics in the dairy herd will inevitably result in an increase in Jersey genetics entering the beef herd from retention of dairy-origin calves for finishing. Limited literature is available on the effect of dam breed on the performance of beef-cross-dairy-breed progeny. The aim of this study was to investigate the effect of dam breed from dams with varying proportions of Friesian and Jersey genetics on growth traits and carcass characteristics of their 24-month-old beef-cross-dairy-breed heifer and steer progeny. Liveweights of 142 heifers and 203 steers from Friesian (F), Friesian-cross (FX), Friesian–Jersey (FJ) and Jersey-cross (JX) dams were recorded at birth, weaning, as yearlings and at slaughter. Carcass characteristics were also recorded. At each point measured, liveweight was greatest for calves born to F dams. Calves born to F dams took 93 days to reach a weaning weight of 100 kg, whereas those from FX, FJ and JX dams took 99, 101 and 102 days, respectively. Carcass weight was greatest for progeny of F dams (286 kg, compared with 279, 275 and 276 for progeny of FX, FJ and JX dams, respectively). The progeny of JX dams had yellower fat than all other dam breed groups and a greater incidence of excessively yellow fat (fat score ≥ 5).

## 1. Introduction

The breed composition of New Zealand dairy herds has shifted from 57% Friesian, 16% Jersey and 19% Friesian–Jersey crossbred in 1998 to 33% Friesian, 8% Jersey and 49% Friesian–Jersey crossbred in 2020 [1,2]. This equates to over a two-fold increase in the proportion of Friesian–Jersey crossbred cows in the dairy herd in the last 23 years to benefit from increased milk production through hybrid vigour. Two-thirds of the annual beef kill in New Zealand originates from the dairy industry (per head), so the increase in crossbred cows in the national dairy herd inevitably results in an increase in Jersey genetics entering the beef herd from retention of dairy-origin calves for beef finishing [3,4].

Calf rearers and beef finishers prefer calves with Friesian markings to those with Jersey markings as cattle with Jersey genetics are expected to have poor growth rates and greater incidences of excessively yellow carcass fat [5]. In the New Zealand carcass classification system, yellow fat can incur a price penalty. Previous literature has reported progeny of Jersey dams to be slower growing than those of Friesian dams, and as the proportion of Jersey genetics increases, dams produce smaller and slower growing progeny with lighter carcasses [5,6,7,8,9,10]. Straightbred Jersey cattle have yellower fat than Friesian cattle, although yellowing was not an issue in beef-cross-dairy-breed steers with only a quarter Jersey genetics [11,12].

There is a widely held belief that animals of dairy origin are inferior beef animals compared to the traditional British beef breeds. This is not supported by the scientific literature. Over 60 years of research confirms there is no difference in growth potential, saleable meat yield, primal cut yield or meat quality between New Zealand dairy breeds and British beef breeds under similar management [13]. In terms of carcass composition, Jersey cattle are more similar to British beef breeds than Friesian cattle, as they have similar proportions of bone, muscle and fat [13]. However, as the percentage of Jersey genetics in the national dairy herd increases, the potential for excessively yellow fat from Jersey genetics of the dam could be a disadvantage for farmers.

The objective of this study was to investigate the effect of dam breed from dams with varying proportions of Friesian and Jersey genetics on growth traits and carcass characteristics of their 24-month-old beef-cross-dairy-breed heifer and steer progeny in a pastoral-based system.

## 2. Materials and Methods

### 2.1. Animals, Management and Slaughter

This experiment included beef-cross-dairy breed calves (145 heifers and 203 steers) born in the spring of 2018. The calves were sired by 26 bulls of 8 beef breeds, and the dams were cows from a New Zealand commercial dairy herd near Taupo, New Zealand, as part of a progeny test of beef bulls over dairy cows. All dams had pedigree confirmed by DNA parentage analysis on a 50 K SNP panel (Neogen, Gatton, Australia) and had their breed proportions known expressed in fractions of 16 where a Friesian dam was considered to have at least 14/16ths Friesian and, likewise, Jersey dams 14/16ths Jersey breed. Using this classification, dams were put into one of four breed groups based on breed composition (Table 1). Cows were bred to one of the 26 bulls using artificial insemination over a 5-week mating period. Sires were randomly assigned to cows based on day of submission for mating, with 5–7 sires used each day of mating.

Calves were collected from their dams within 24 h of birth and artificially reared at one of two sites within the same farming complex. Parentage was assigned using DNA, using a 6.5 K SNP panel (Genomz, Dunedin, New Zealand). Male calves were castrated before weaning. Calves were weighed every 2 weeks as they approached weaning weight and were weaned once they reached a minimum weight of 85 kg. After weaning, calves were moved to a single finishing farm near Taupo, New Zealand.

In January 2019, the calves were drafted into 4 management groups based on sex and live weight within sire group: heavy heifers, light heifers, heavy steers and light steers. They remained in these groups for the remainder of the study. Cattle were grazed under commercial management until processing at a target of 500 kg live weight for heifers and 600 kg live weight for steers. Each herd, apart from the heavy heifers, was drafted into two sub-herds based on live weight within sire group in the lead-up to slaughter, and all cattle within a sub-herd were processed on a single day, to give 7 processing events. Processing was conducted according to standard commercial practices at Silver Fern Farms’ Pacific plant in Hawkes Bay, New Zealand.

### 2.2. Measurement of Live Animals

Calves were weighed on arrival at the calf shed where they were to be reared, prior to being fed, as a measure of birth weight. Date of birth was recorded as the date of arrival at the calf shed, and gestation length was calculated from the mating records for the dam and sire of the calf. Unfasted live weight was recorded at weaning, as yearlings and prior to slaughter. The age of animals in days was also recorded at weaning, as yearlings and at slaughter.

### 2.3. Measurement of Carcass Characteristics

Immediately after dressing, carcasses were weighed to give a hot carcass weight. The dressing-out percentage was calculated as the ratio of hot carcass weight to preslaughter liveweight (taken on farm) and expressed as a percentage. The carcasses were graded using the Silver Fern Farms’ Eating Quality (EQ) System^®^ by an accredited master grader. Graders visually assessed fat colour, meat colour, marbling and ossification against a set of reference standards [14]. Fat colour was scored on a scale from 0 (white) to 9 (deep yellow), whereas meat colour was scored on a scale from 1 (light red) to 7 (dark red); ossification was scored on a scale from 100 (no ossification) to 590 (complete fusion of sacral vertebrae, complete ossification of lumbar vertebrae and thoracic vertebrae outlines barely visible), and marbling was scored on a scale from 100 to 1190 [14]. Fat depth and meat pH were measured using a ruler and a pH meter, respectively.

### 2.4. Statistical Analysis

Statistical analysis was conducted using Statistical Analysis System software version 9.4 (SAS institute Inc., Carey, NC, USA) to obtain the least squared means for dam breed. All analyses included the random effect of sire because the focus of this research was to consider dam breed effects. Therefore, the variation in the measured growth and carcass characteristics associated from the sire was accounted for by incorporating sire into the statistical models as a random effect.

Birth weight was analysed with the fixed effect of dam breed and the covariate effect of sex. Analysis of weaning age included the fixed effect of dam breed and covariate effects of sex and rearing site. Yearling weight was analysed with the fixed effect of dam breed and the covariate effects of rearing site and management group. The analysis of slaughter weight included the fixed effect dam breed and covariate effect of management group. Average daily gain was analysed using dam breed as the fixed effect and rearing site (preweaning) or management group (weaning to slaughter) as a covariate effect.

The analysis for the carcass characteristics included the fixed effect of dam breed and covariate effect of slaughter group. Additionally, fat depth and marbling score analyses included carcass weight as a covariate, meat colour analyses included pH as a covariate and ossification score analysis included age at slaughter as a covariate.

The proportion of carcasses that had a fat colour score greater than or equal to 5 were analysed using a generalised linear model based on a binomial distribution.

The percentage of cattle with a fat colour score greater than or equal to 5 for each breed was used to calculate the impact of dam breed on value of the cattle at processing, assuming a 27% reduction in price per kg for cattle downgraded from P2 to M (NZD 5.28/kg for cattle graded P2, and NZD 3.83/kg for cattle graded M). Prices were taken as the average of the North Island processor prices from 29/10/2020 to 30/03/2021, as reported by AgriHQ (GlobalHQ, Fielding, New Zealand).

## 3. Results

### 3.1. Growth Traits

Calves born to Friesian–Jersey and Jersey-cross dams had lower birth weights than calves born to Friesian dams (*p* < 0.05, Table 2). Calves born to Friesian dams took less time to reach weaning weight compared with calves born to the other dam breeds. There was no difference in time till weaning weight was reached between calves born to Jersey-cross dams and those born to Friesian-cross and Friesian–Jersey dams.

Calves born to Friesian dams had 7 kg greater yearling weight than those born to Friesian–Jersey dams. Average daily gain from birth to weaning and from weaning to yearling age was not affected by dam breed group (*p* > 0.05, Table 3). From yearling age to slaughter, however, progeny of Friesian dams had a greater average daily gain than progeny from Friesian–Jersey dams.

### 3.2. Carcass Characteristics

Carcass weight was heaviest for progeny of Friesian dams and lightest for progeny of Friesian–Jersey dams with Friesian-cross progeny intermediate (Table 4). There was no difference in carcass weights between progeny of Jersey-cross dams compared to progeny from Friesian-cross and Friesian–Jersey dams. Progeny of Jersey-cross dams had greater fat colour scores and a greater frequency of excessively yellow fat (≥5) than the progeny from other dam breeds. There was no difference in fat colour between progeny of Friesian, Friesian-cross and Friesian–Jersey dams. Ossification score was highest for progeny from Jersey cross dams and lowest for progeny from Friesian–Jersey dams, with progeny from Friesian and Friesian cross having an intermediate ossification score. Fat depth was greatest for progeny from Friesian dams and lowest for progeny from Friesian-Jersey dams, with progeny from Friesian cross and Jersey cross dams having an intermediate fat depth. There was no difference in fat depths between progeny from Jersey cross and Friesian–Jersey dams, however. There was no difference in dressing-out percentage, marbling score, meat colour score or pH between dam breed groups. The mean reduction in value per kg of carcass weight that would result if carcasses with fat colour scores of five or more were downgraded to manufacturing grade was 0.7% for progeny of Friesian dams, 1.5% for progeny from Friesian-cross dams, 2.5% for progeny from Friesian–Jersey dams and 4.7% for progeny from Jersey-cross dams.

## 4. Discussion

It is well-documented that Friesian cows produce heavier calves at birth compared to other dairy breeds including Jersey, Ayrshire and Guernsey [3,7,15,16,17]. This is likely a reflection of Friesian cattle having a greater mature size than the other dairy breeds [18]. Birth weight was lowest for calves born to Friesian–Jersey and Jersey-cross dams, whereas calves born to Friesian–cross dams had intermediate birthweights. This is consistent with previous literature, as calves born to crossbred dairy cows tend to have intermediate birthweights compared to those born to straightbred dairy cows [7].

Weaning age was affected by dam breed group, with progeny from Friesian dams reaching weaning earlier than all other dam breed groups. This is often attributed to differences in growth rates, as it is well-documented that Friesian cattle grow faster than Jersey cattle so would reach a set weaning weight earlier [18,19]. In the current study, although no difference in growth rates between dam breed groups was observed between birth and weaning, growth rates of Friesian progeny were numerically greater, which would have affected time until weaning weight was reached. In addition, the calves were heavier at birth, and so required less live weight to gain to reach weaning weight than calves of other breeds. A scaled weaning weight could have been more relevant as calf rearers typically either wean once a certain increase in liveweight has been achieved or have separate weaning weight targets for different breed calves. It is recommended that calves artificially reared for beef production gain at least 18 kg liveweight before weaning, and the target weaning weight for dairy heifers is 100 kg, 90 kg and 80 kg for Friesian, crossbred and Jersey calves, respectively [20,21,22,23].

The greater yearling weight of progeny from Friesian dams is consistent with previous literature, in which Friesian or Friesian-cross steers were heavier at 12–13 months compared with Jersey and Jersey-cross steers [7,19]. Yearling weights were approximately 100 kg and 50 kg lower than those reported by Pike [24] and Coleman [25] for beef-cross-dairy-breed steers and heifers, respectively. This suggests that the growth of the animals in this study was restricted relative to previous studies, which could explain why less of a difference was observed between breed groups, as the animals were unable to exhibit their full potential for growth. Changeable feed availability and environmental conditions over the course of the study would have restricted animal growth.

Cattle with Friesian genetics have heavier carcass weights compared with cattle with Jersey or Friesian–Jersey genetics [6,18,25,26], and this was observed in this experiment. Although progeny of Friesian–Jersey dams had the lightest carcass weights, they did not differ from progeny of Jersey-cross dams. This is in contrast with Barton et al. [19] who found Friesian–Jersey steers to have heavier carcasses than Jersey steers, although these animals were straightbred so would have had twice the amount of Friesian genetics than the animals in the current experiment. Coleman [25] found no difference in carcass weights between beef-cross-dairy-breed progeny of dams with Jersey and Friesian–Jersey genetics. At NZD 5.28/kg carcass weight, the 13 kg greater carcass weight of progeny from Friesian dams compared with progeny from Friesian-Jersey dams equates to an NZD 68.64 difference in sale value, or a 4.5% greater carcass value for the progeny from Friesian dams.

Dam breed group had no effect on dressing out percentage. Although this is consistent with Coleman [25] and Barton et al. [27], cattle with Jersey genetics do have lower dressing out percentages [6,18,25,26]. This was attributed to Jersey steers depositing greater amounts of fat in non-carcass components relative to Friesian steers, particularly around the kidneys [28,29]. Dam breed group did have an influence on fat depths, which is inconsistent with previous literature reporting no difference in fat depth between straight Friesian and Jersey cattle, and between beef-cross-dairy-breed cattle from Friesian, Jersey and Friesian–Jersey dams [18,24,25]. Progeny from all dam breeds had fat depths within the same fat class range for carcass classification, however, meaning dam breed differences in fat depth had no effect on carcass grading.

Progeny of Jersey-cross dams had yellower fat colour scores than progeny from all other breed groups. It is well-documented that Jersey cattle and cattle with Jersey genetics have yellower fat than cattle with Friesian genetics [11,30]. The yellowing of fat is due to carotenoid pigments in the diet, which are typically converted into Vitamin A before absorption [30]. Jersey cattle lack the enzyme required to catalyse this reaction, and carotenes are absorbed into the bloodstream and deposited in adipose tissue as they are lipophilic compounds [31]. Numerically, the difference between mean fat colour scores was less than 0.5. Although statistically significant, this difference may be too small to be detectable by consumers. Nevertheless, when considering the number of carcasses with a fat colour score ≥ 5, this translated to a 4.7% reduction in value per kg carcass for progeny of Jersey-cross dams. This difference in value of the carcass should be considered by beef finishers when determining the likely return on cattle purchased for their finishing system.

The greater incidence of excessive yellowness in the fat of Jersey-cross cattle can be partially overcome by different management of cattle with Jersey genetics. Levels of carotenes in pasture vary between seasons [32,33]. This is associated with the effect of climactic conditions on the stage of growth and maturity of pastures [33,34]. β-carotene and xanthophyll concentrations are greater when ryegrass is immature, between late spring to early summer pastures, and results in greater blood and adipose concentrations of carotene in cattle over this period [33]. This is consistent with the current study; 24% of the first steer group sent to slaughter in early November had excessively yellow fat, whereas none of the steers sent to slaughter in late January had excessively yellow fat. To reduce the loss from yellow fat, farmers could send beef-cross-dairy-breed cattle with Jersey genetics to slaughter in mid to late summer.

Dam breed group did not affect marbling score in this study, which is consistent with previous studies comparing marbling scores and intramuscular fat content of beef-cross-dairy-breed cattle from dams with Friesian, Jersey and Friesian–Jersey genetics [25,34,35]. Previous literature on straightbred Jersey and Friesian steers has found Jersey steers to have greater potential for deposition of intramuscular fat as a result of being earlier maturing than Friesians [35]. This may not be observed in the crossbred progeny from this and previous experiments as they contained a percentage of Jersey genetics of less than 50%.

Dam breed group had no effect on meat pH, and this has been observed previously [35,36]. Variation in pH among cattle is usually explained by differences in pre- and post-slaughter treatments, as opposed to breed or genotype [36]. As with pH, there was no difference in meat colour between dam breed groups. Lean meat colour is highly correlated with pH and, as a result, is affected mostly by management of cattle prior to slaughter [37]. Although differences in lean colour between breed types (e.g., dairy breeds, Continental beef breeds and British beef breeds) have been found, lean meat colour is typically similar between breeds within a breed type [37].

Ossification score is affected by physiological maturity rather than breed [38,39]. Differences among breed groups in ossification score may be explained by differences in degree of maturity of the progeny among the breeds. Compared to Friesian cattle, Jersey cattle mature earlier [25]. The greater ossification scores observed in progeny of Jersey-cross dams could therefore be explained by the greater proportion of earlier maturing genetics in these animals [38]. Actual ossification scores for progeny of all breed groups were lower than those reported by Bonny et al. [38] for beef-breed cattle of a similar age but were similar to the standard ossification scores of the Meat Standards Australia grading system for 24-month-old cattle [14].

## 5. Conclusions

Progeny from Friesian dams had faster growth than progeny from other dam breed groups, taking less time to reach weaning and having heavier birth, yearling and slaughter weights compared to other dam breed groups. Although growth rates were similar among dam breed groups up until yearling age, Friesian progeny had faster growth rates than Friesian–Jersey progeny from yearling to slaughter. Carcass characteristics were less affected by dam breed than growth traits, with carcass weight, fat colour and ossification score being the only carcass characteristics influenced by dam breed. Carcass weight and fat colour, however, are particularly important characteristics to beef producers, as they determine the return to the farmer. Compared with progeny from Friesian dams, carcasses from the various crossbred dams were 2.4–4.5% lighter and valued at 0.8–4.1% less per kg as a result of increased incidence of yellow fat, and this will likely reduce the price that finishers will pay for beef-cross calves from crossbred dairy dams.

## Figures and Tables

**Table 1 animals-12-00864-t001:** Breed composition and number of cows for Friesian (F), Friesian crossbred (FX), Friesian-Jersey crossbred (FJ) and Jersey crossbred (JX).

Breed Group	Breed Composition	*n*
F	F ≥ 14/16	40
FX	10/16 ≤ F ≤ 13/16	113
FJ	F < 10/16 and J < 10/16	148
JX	10/16 ≤ J ≤ 13/16	44

**Table 2 animals-12-00864-t002:** Growth traits for beef-cross-dairy-breed progeny from Friesian (F), Friesian-cross (FX), Friesian–Jersey (FJ) and Jersey-cross (JX) dams. Values are least squared means, ±standard error of the mean.

	Dam Breed				
Trait	F	FX	FJ	JX	*p*-Value
*n*	40	113	148	44	-
Birthweight (kg)	41.5 ± 0.77 ^a^	38.0 ± 0.47 ^b^	36.8 ± 0.44 ^c^	36.8 ± 0.72 ^bc^	<0.001
Weaning age (d)	92.8 ± 2.31 ^a^	98.6 ± 1.36 ^b^	101.1± 1.23 ^b^	102.3 ± 2.13 ^b^	<0.05
Yearling weight (kg)	225 ± 3.36 ^a^	221 ± 1.98 ^ab^	218 ± 1.81 ^b^	219 ± 3.09 ^ab^	<0.001
Slaughter weight (kg)	594 ± 5.06 ^a^	580 ± 2.94 ^b^	571 ± 2.70 ^c^	574 ± 4.72 ^bc^	<0.001

^a–c^ Values within row without superscripts in common differ at the *p* < 0.05 level.

**Table 3 animals-12-00864-t003:** Average daily gain (kg/day) throughout the study of beef-cross-dairy-breed progeny from Friesian (F), Friesian-cross (FX), Friesian–Jersey (FJ) and Jersey-cross (JX) dams. Values are least squared means, including the standard error of the mean.

	Dam Breed				
Time Period	F	FX	FJ	JX	*p*-Value
Preweaning	0.665 ± 0.01	0.650 ± 0.01	0.646 ± 0.01	0.652 ± 0.01	0.341
Weaning-Yearling	0.519 ± 0.01	0.524 ± 0.01	0.515 ± 0.01	0.519 ± 0.01	0.125
Yearling-Slaughter	0.680 ± 0.1 ^a^	0.655 ± 0.01 ^ab^	0.643 ± 0.01 ^b^	0.650 ± 0.01 ^ab^	0.002

^a,b^ Values within row without superscripts in common differ at the *p* < 0.05 level.

**Table 4 animals-12-00864-t004:** Carcass characteristics for beef-cross-dairy-breed progeny from Friesian (F), Friesian-cross (FX), Friesian–Jersey (FJ) and Jersey-cross (JX) dams. Values are least squared means, ± standard error of the mean.

	Dam Breed				
Trait	F	FX	FJ	JX	*p*-Value
*n*	38	107	141	41	-
Carcass weight	286 ± 2.79 ^a^	279 ± 1.62 ^b^	273 ± 1.49 ^c^	276 ± 2.58 ^bc^	<0.001
Dressing out (%)	48 ± 0.002	48 ± 0.001	48 ± 0.001	48 ± 0.002	0.309
Fat colour score	2.85 ± 0.15 ^a^	3.04 ± 0.09 ^a^	3.05 ± 0.8 ^a^	3.33 ± 0.14 ^b^	<0.05
Fat colour score ≥ 5 (%) ^1^	2.6 (1.9,3.27)	5.6 (4.9,6.3)	9.2 (8.5,9.9)	17.1 (16.4,17.8)	-
Fat depth (mm)	5.94 ± 5.06 ^a^	5.80 ± 2.94 ^b^	5.71 ± 2.70 ^c^	5.74 ± 4.72 ^bc^	<0.001
Marbling score ^2^	316 ± 18	319 ± 10	318 ± 9	302 ± 16	0.342
Meat colour score ^3^	3.41 ± 0.11	3.64 ± 0.06	3.59 ± 0.06	3.68 ± 0.10	0.683
pH	5.62 ± 0.02	5.61 ± 0.01	5.59 ± 0.01	5.59 ± 0.02	0.660
Ossification score ^4^	162 ± 2.76 ^ab^	159 ± 1.59 ^ab^	157 ± 1.46 ^a^	164 ± 2.53 ^b^	<0.005

^a–c^ Values with different superscript are significantly different from one another. ^1^ Confidence limits (0.05) are presented instead of standard errors as means are back transformed. ^2^ On a scale from 100 to 1190. ^3^ On a scale from 1 to 7. ^4^ On a scale from 100 to 590.

## Data Availability

Data sharing is not applicable to this article.

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
