# Peer review of "Growth and Carcass Characteristics of Beef-Cross-Dairy-Breed Heifers and Steers Born to Different Dam Breeds"

_animals, 2022, doi:10.3390/ani12070864_

Round 1
Reviewer 1 Report
The manuscript investigated the effect of dam breed from dams with different ratio of Friesian and Jersey genetics on growth traits and carcass characteristics. The topic of the work is interesting, however, the manuscript needs to be edited to improve readability and add depth. My main concern is related to statistical analysis. This section is not accurate, statistical models is not clear. I invite the authors to clarify the model and the fixed effects and to specify the type of analysis used.
Several parts of results section are not accurate which need to be further improved.
Specific comments:
Line 119: live weight is not in the table…it is birth weight? please clarify.
In the result section please don't repeat tables number each time but first introduce the table and after describe the results.
Table 2 Please delete from weaning until slaughter
Reviewer 2 Report
This is an area of timely and increasing interest globally. As a result, this work should be relevant to the global scientific community. The paper is generally well-written, although minor sentence structure and grammatical errors should be improved before this manuscript is published. This reviewer has minor suggested edits and comments below.
Line 16: This is a long sentence that I would suggest splitting into two separate sentences where the comma currently resides.
Line 23: "recorded" in place of "taken" seems more correct.
Line 43: Suggested to revise to ". . . excessively yellow carcass fat."
Line 50: Suggested revision to ". . . beef animals compared to the traditional . . ."
Statistical Analysis: Were there any effects due to sire breed that should be discussed?
Line 168: Suggested revision to "The mean reduction in value per kg of carcass weight"
Table 4: Please check the decimal place and units for the fat depth values. A range of 574-594 mm seems very high. Should this be 5.74-5.94 mm?
Regarding the carcass traits presented, were data collected for ribeye area or other indicators of red meat yield?
Round 2
Reviewer 1 Report
Even if the paper has been improved, the statistical models is not still clear
How many fixed effects were tested? On the basis of all tables I understand that only the fixed effect of dam breed has been tested
What do you mean that the fixed effects included in the analysis were stated for each variable separately?
Minor suggestion
Line 119 please change weight in weights
Line 121 please change effects in effect
